# Univariate vs Multivariate Time Series Forecasting with Transformers

## Abstract

Multivariate time series forecasting is a challenging problem and a number of Transformer-based long-term time series forecasting models have been developed to tackle it. These models, however, are impeded by the additional information available in multivariate forecasting. In this paper we propose a simple univariate setting as an alternative method for producing multivariate forecasts. The univariate model is trained on each individual dimension of the time series. This single model is then used to forecast each dimension of the multivariate forecast in turn. A comparative study shows that our setting outperforms state-of-the-art Transformers in the multivariate setting in benchmark datasets. To investigate why, we set three hypotheses and verify them via an empirical study, which leads to a criterion for when our univariate setting is likely to lead to better performance and reveals flaws in the current multivariate Transformers for long-term time series forecasting.

## 1 Introduction

In an ever increasingly digital world, technology and the cost of data collection is becoming cheaper. Time series are being generated of greater lengths and dimensionality, and so more is being demanded of time series forecasting (TSF). Their application ranges across industry, such as electricity forecasting, stock prediction, and health, (Torres et al., 2021) and so any improvements in TSF can have far reaching benefits for society.

The advent of deep learning brought with it multiple different TSF architectures and saw new ground continuously broken with models such as the recurrent neural network (RNN) (Hochreiter & Schmidhuber, 1997), temporal convolutional networks (TCNs) (Bai et al., 2018), and attention (Vaswani et al., 2017).

Attention based models offer the unique benefit that the path length between distant dependencies is always one. This property has seen models, such as the Transformer, outperform previous state of the art (SOTA) methods by a significant margin in fields such as natural language processing (NLP) (Brown et al., 2020) and computer vision (Dosovitskiy et al., 2020). While the Transformer performs very well in TSF, it suffers from $O(l^2)$ complexity, where $l$ is the length of the input to the model.

Predictive patterns can be found across distant time steps and so increasing the length of the input to the model has been found to improve accuracy (Li et al., 2019). The field of research for efficient Transformers for long-term TSF has therefore been an important new frontier.

Multiple efficient Transformers have been developed for TSF such as the Informer by Zhou et al. (2021) and the Autoformer by Xu et al. (2021) which both achieve a complexity of $O(l \cdot log(l))$. The FEDformer, by Zhou et al. (2022), manages to achieves linear complexity.

The authors evaluate each of these models on the same pool of benchmark datasets, in both a multivariate and univariate setting. While this is valuable in determining which model achieves the best forecasting accuracy, the method by which their univariate mode is implementing means that comparisons between the two settings cannot be made.

In the multivariate mode, multivariate sequences of dimension $d$ are inputted and all of the $d$ dimensions are forecasted in a single multivariate output. The reported loss is the average loss of

the forecast over every dimension. Their univariate setting, however, involves selecting a single dimension from each dataset, which the model is trained and tested on. Since only one dimension is forecasted, the results cannot be compared to the multivariate results which are comprised of all dimensions.

We alter the univariate mode, enabling it to be compared with its multivariate counterpart. We train a single univariate model to forecast every dimension individually. These separate forecasts can be combined to recreate the multivariate forecast. In doing so, we find that our univariate setting produces more accurate forecasts than the multivariate setting, despite its limitation of only being able to operate on a single dimensions at a time.

The finding is surprising since the intra-dimensional patterns that the univariate model learns are still present in the multivariate setting. The multivariate model, in theory, should be able to learn these, as well as inter-dimensional patterns.

This then raises two questions. Whether the multivariate datasets contain predictive inter-dimensional patterns for the models to learn. And secondly, why the multivariate models are unable to learn the same intra-dimensional patterns that univariate models find.

Via empirical study, we address these two issues and in doing so devise guidelines for when a multivariate model is likely to be more appropriate than combining the results of a univariate one. Our work should provide valuable insight into improving Transformer based long-term TSF models for the multivariate setting.

## 2 METHOD

We compare the multivariate, and our univariate setting across four transformer based TSF models and find that in the majority of cases, our univariate setting achieves SOTA results.

Multivariate models, in theory, should be able to match or outperform their counterpart since they have access to the same information, plus more. We address three questions that this raises: **(1)** - There are no inter-dimensional patterns for the multivariate models to learn. **(2)** - The multivariate models are impeded by the additional dimensions (the dimension count). **(3)** - Models in the multivariate setting require more training data to outperform the univariate mode.

### 2.1 MODELS

We evaluate our univariate setting and its multivariate counterpart with four models. Three of the models are SOTA Transformer-based long-term TSF, such as the Informer, by Zhou et al. (2021), which achieves a computational complexity of $O(l \cdot log(l))$ via its *ProbSparse* self-attention mechanism, only allowing each key to attend to the most dominant queries. The Autoformer, by Xu et al. (2021), likewise achieves $O(l \cdot log(l))$ with its *Auto-Correlation* mechanism which discovers period-based dependencies. The FEDformer, by Zhou et al. (2022), achieves linear complexity with its *Fourier enhanced structure* and by representing the series in the time domain and randomly selecting a constant number of Fourier components.

Additionally, we test with a Vanilla Transformer (VT) to examine what advantages its $O(l^2)$ complexity bring. Including this model will also help us to generalise our findings to all Transformer-based TSF architectures. We aim to keep the model as simple as possible. The time series are embedded with a 1-D convolutional neural network (CNN) of kernel size, 7. The output of the single layered transformer is projected to the horizon length $h$ with a fully connected layer. The exact implementation of the models can be found in our repository `www.github.com`.

### 2.2 BENCHMARK DATASETS

We evaluate the models on all the standard benchmark datasets that have been used by previous long-term TSF models such as the Informer, Autoformer and the FEDformer.

The datasets are, Electricity, which contains hourly usage measurements of 321 customers from 2012-2015. Electricity Transformer Temperature (ETT), collected from 2016-2018 in China, which includes 7 features/dimensions of the load and oil temperatures of two stations. Of the 4 variations

we use the *M2*. Exchange, the daily exchange rates of 8 countries between 1990 to 2016. Influencza-like illness (ILI), the weekly number of patients in the United states from 2002 to 2021, which includes 7 features. Traffic, 862 dimensions of hourly lane occupancy rates in California between 2015-2016. Weather, which contains 21 meteorological indicators for the year of 2020. Table 1 gives a useful summary of the dataset statistics.

| Datasets | Features | Length |
|---|---|---|
| Electricity | 321 | 26304 |
| ETT | 7 | 69680 |
| Exchange | 8 | 7588 |
| ILI | 7 | 966 |
| Traffic | 862 | 17544 |
| Weather | 21 | 52696 |

Table 1: Statistics of the 6 benchmark datasets.

## 2.3 UNIVARIATE FORECASTING

Recent SOTA long-term TSF models, Informer, Autoformer and the FEDformer are evaluated on a range of datasets in both the univariate and multivariate setting, however, due to differences between the methods, comparisons between the two cannot be made.

In the multivariate setting, multivariate sequences are both inputted and outputted from the model, $M_m$. The reported loss is the average loss over the entire forecast over every dimension. However, the univariate setting involves selecting a single dimension from each dataset, which the model is trained and tested on. Since only one dimension is forecasted, the results cannot be compared to the multivariate results which are comprised of all dimensions. Therefore we alter the univariate setting to produce a multivariate output.

Traditionally, to obtain a multivariate forecast, $\hat{X}_t$ with univariate models, $M_t$, each dimension of the input, $X$, would be separated, Eq (1), and a different model applied to each dimension, Eq (2). This requires $d$ separate models to be trained, incurring a significant computational cost for high dimensional datasets. The Traffic dataset we use has 862 dimensions and so 862 models would be required.

Therefore, we train a single univariate model, $M_u$, on all the individual dimensions. During training, this involves repeatedly taking sequences from a random dimension to feed into the model. The end result is a model that can forecast any single dimension. To produce the multivariate forecast, $\hat{X}_u$, all the univariate forecasts are concatenated together, see Eq (3).

For the multivariate model, $M_m$, the input, $X$, is not separated and the entire multivariate forecast, $\hat{X}_m$, is produced in one step, see Eq (4).

With our univariate setting, the forecast $\hat{X}_u$ can be directly compared with the forecast $\hat{X}_m$ from the multivariate model, and does not require the computational cost of producing $\hat{X}_t$.

$$X = x_1, x_2, \cdots, x_d \tag{1}$$

$$\hat{X}_t = M_{t_1}(x_1), M_{t_2}(x_2), \cdots, M_{t_d}(x_d) \tag{2}$$

$$\hat{X}_u = M_u(x_1), M_u(x_2), \cdots, M_u(x_d) \tag{3}$$

$$\hat{X}_m = M_m(X) \tag{4}$$

## 2.4 INTER-DIMENSIONAL DEPENDENCIES

Multivariate models have the potential to outperform univariate models since they have access to more information. Univariate models only operate over single dimensions, limiting the model to learning intra-dimensional patterns. Multivariate models operate over multiple dimensions and so can learn inter-dimensional patterns, in addition to the intra-dimensional kind.

We therefore want to understand how the multivariate models are making the forecast and whether they are utilising inter-dimensional patterns. If the benchmark datasets contain predictive inter-dimensional patterns, then the multivariate models should outperform their univariate counterpart.

Attention based models can often lend themselves to interpretation via their attention weights. The query key dot products show which inputs are amplified and therefore of high predictive relevance. This method, however, is not realisable due to the use of one dimensional CNNs being used to embed the input time sequence. The kernel of the CNN sums all the dimensions together before attention is applied, obfuscating direct access to the dimensions.

Instead, to find the strength of inter-dimensional dependencies we look at what impact each input dimension has on the forecasted dimensions. If one dimension contributes to the forecast of multiple output dimensions, then the model has likely learnt a predictive pattern between input dimensions. A pattern that could not be identified with a univariate model.

We achieve this by computing the Jacobian matrix, $J$, of the output. For every forecasted dimension within every time step of the output, $\hat{X}$, we compute the partial derivative with respect to the input $X$,

$$
J = \begin{bmatrix} \frac{\partial \hat{X}_{11}}{\partial X} & \cdots & \frac{\partial \hat{X}_{1h}}{\partial X} \\ \vdots & \ddots & \vdots \\ \frac{\partial \hat{X}_{d1}}{\partial X} & \cdots & \frac{\partial \hat{X}_{dh}}{\partial X} \end{bmatrix}
\tag{5}
$$

We are interested in the overall affect of each input dimension with the output dimensions, and so we simplify $J$ by taking the average along the time step axis. This reduces the size of $J$ from $d \times h$ to $d$, where $h$ is the forecast horizon,

$$
\bar{J}_i = \frac{1}{h} \sum_{k=0}^{h} |J_{i,k}|
\tag{6}
$$

Each element within $\bar{J}$ is still of size $d \times l$, where $l$ is the length of the input, and so we summarise this further by averaging all the time steps,

$$
D = \frac{1}{l} \sum_{i=0}^{l} |\bar{J}_i|
\tag{7}
$$

The matrix $D \in \mathbb{R}^{d \times d}$ shows the overall influence each input dimensions has over each output dimension. Input dimensions that have a high impact on certain output dimensions, can be interpreted as having high predictive utility for those specific outputs.

From $D$ we are able to tell whether inter-dimensional patterns have been learnt. If no such patterns have been found, $D$ should be similar to the identity matrix. If the patterns are present then input dimensions should affect multiple output dimensions, see Figure 1.

## 2.5 Dimension count impact

To better understand the interplay between the number of dimensions a dataset has and the performance of the trained model, we create an array of new datasets. These are subsets of the benchmark datasets with dimensions removed.

For example, the Electricity dataset contains 321 dimensions, see Table 1. We take $r$ randomly selected dimensions, maintaining the series length $l$, to create the new subdataset. This is then used to train and test the model. By repeating this process with a range of values of $r$ we are able to create a plot of how the number of dimensions affects the accuracy of the multivariate model.

When $r$ is equal to the full number of dimensions within the dataset, $d$, the case generalises to the full multivariate setting. When $r$ is equal to 1 we reach the univariate setting. This experiment will give a clear view on how additional dimensions impact the performance of a model.

## 2.6 DATASET SIZE REQUIREMENTS

Often in machine learning, if a model performs poorly, a lack of training data can be blamed. And so we test the data requirements for both the univariate and multivariate setting.

While it is challenging if not impossible to increase the size of a dataset, we can easily reduce the size by removing time steps. If it is expected that increasing the size would improve the performance, than it should hold that reducing it would worsen the performance.

Therefore, we create an array of subdatasets with reduced training set sizes. The validation and test splits remain unchanged. When time steps are removed, they are removed in chronological order starting at the beginning of the series. Training and testing a model on all the subdatasets, in both the univariate and multivariate setting, shows how sensitive each mode is to the amount of data.

| | History | Horizon | VT Univariate MSE | MAE | VT Multivariate MSE | MAE | Informer Univariate MSE | MAE | Informer Multivariate MSE | MAE | Autoformer Univariate MSE | MAE | Autoformer Multivariate MSE | MAE | FEDformer Univariate MSE | MAE | FEDformer Multivariate MSE | MAE |
|---|---|---|---|---|---|---|---|---|---|---|---|---|---|---|---|---|---|---|
| Electricity | 96 | 96 | 0.194 | 0.280 | 0.311 | 0.410 | **0.154** | **0.248** | 0.282 | 0.377 | 0.196 | 0.288 | 0.202 | 0.316 | 0.213 | 0.300 | 0.196 | 0.312 |
| | 96 | 192 | 0.194 | 0.283 | 0.312 | 0.412 | **0.169** | **0.262** | 0.276 | 0.368 | 0.227 | 0.309 | 0.220 | 0.327 | 0.216 | 0.303 | 0.205 | 0.320 |
| | 96 | 336 | 0.207 | 0.297 | 0.316 | 0.414 | **0.196** | **0.293** | 0.285 | 0.376 | 0.237 | 0.321 | 0.224 | 0.334 | 0.224 | 0.317 | 0.216 | 0.331 |
| | 96 | 720 | 0.239 | 0.328 | 0.326 | 0.418 | **0.226** | **0.317** | 0.309 | 0.391 | 0.379 | 0.420 | 0.263 | 0.360 | 0.259 | 0.343 | 0.239 | 0.348 |
| ETT | 96 | 96 | **0.114** | **0.232** | 0.197 | 0.343 | 0.128 | 0.246 | 0.179 | 0.297 | 0.137 | 0.261 | 0.139 | 0.255 | 0.126 | 0.252 | 0.124 | 0.243 |
| | 96 | 192 | **0.143** | **0.264** | 0.225 | 0.340 | 0.158 | 0.280 | 0.221 | 0.336 | 0.169 | 0.292 | 0.192 | 0.297 | 0.159 | 0.282 | 0.150 | 0.267 |
| | 96 | 336 | **0.178** | 0.300 | 0.364 | 0.417 | 0.198 | 0.312 | 0.239 | 0.347 | 0.198 | 0.313 | 0.207 | 0.313 | 0.193 | 0.307 | 0.183 | **0.294** |
| | 96 | 720 | **0.230** | 0.346 | 0.881 | 0.575 | 0.265 | 0.369 | 0.438 | 0.480 | 0.261 | 0.360 | 0.264 | 0.359 | 0.252 | 0.352 | 0.244 | **0.339** |
| Exchange | 96 | 96 | **0.066** | **0.191** | 0.553 | 0.563 | 0.188 | 0.316 | 0.966 | 0.795 | 0.121 | 0.252 | 0.152 | 0.278 | 0.097 | 0.219 | 0.127 | 0.257 |
| | 96 | 192 | **0.119** | **0.259** | 1.320 | 0.912 | 0.320 | 0.433 | 1.228 | 0.878 | 0.244 | 0.360 | 0.402 | 0.459 | 0.200 | 0.324 | 0.274 | 0.379 |
| | 96 | 336 | **0.194** | **0.331** | 0.857 | 0.754 | 0.653 | 0.617 | 1.508 | 1.005 | 0.469 | 0.509 | 0.649 | 0.598 | 0.358 | 0.438 | 0.440 | 0.490 |
| | 96 | 720 | **0.741** | **0.661** | 1.833 | 1.106 | 1.236 | 0.872 | 2.468 | 1.309 | 2.123 | 1.158 | 1.408 | 0.885 | 0.973 | 0.755 | 1.232 | 0.846 |
| Illness | 36 | 24 | 2.630 | 1.121 | **2.060** | 1.010 | 4.394 | 1.357 | 5.088 | 1.490 | 3.084 | 1.141 | 3.508 | 1.256 | 2.553 | **0.988** | 2.723 | 1.129 |
| | 36 | 36 | 3.114 | 1.284 | **2.113** | 1.012 | 4.392 | 1.362 | 4.746 | 1.444 | 2.828 | 1.081 | 3.348 | 1.187 | 2.532 | **0.968** | 3.028 | 1.187 |
| | 36 | 48 | 3.094 | 1.283 | **2.598** | 1.118 | 4.131 | 1.319 | 4.642 | 1.440 | 2.907 | 1.096 | 3.176 | 1.160 | 2.889 | **1.078** | 2.873 | 1.088 |
| | 36 | 60 | 2.905 | 1.233 | **2.791** | 1.166 | 4.315 | 1.382 | 4.885 | 1.478 | 2.820 | 1.079 | 3.003 | 1.121 | 2.868 | **1.071** | 3.008 | 1.163 |
| Traffic | 96 | 96 | **0.391** | 0.339 | 0.428 | 0.382 | 0.544 | **0.257** | 0.673 | 0.369 | 0.547 | 0.318 | 0.639 | 0.395 | 0.540 | 0.377 | 0.586 | 0.367 |
| | 96 | 192 | **0.372** | 0.315 | 0.431 | 0.388 | 0.491 | **0.260** | 0.721 | 0.387 | 0.616 | 0.339 | 0.675 | 0.423 | 0.659 | 0.381 | 0.606 | 0.374 |
| | 96 | 336 | **0.379** | 0.317 | 0.454 | 0.399 | 0.483 | **0.269** | 0.716 | 0.391 | 0.590 | 0.335 | 0.634 | 0.391 | 0.659 | 0.377 | 0.624 | 0.387 |
| | 96 | 720 | **0.402** | 0.337 | 0.450 | 0.385 | 0.611 | **0.305** | 0.796 | 0.424 | 0.688 | 0.374 | 0.652 | 0.398 | 0.637 | 0.391 | 0.624 | 0.378 |
| Weather | 96 | 96 | 0.194 | **0.257** | 0.189 | 0.281 | 0.222 | 0.306 | 0.338 | 0.396 | 0.223 | 0.293 | 0.228 | 0.305 | 0.217 | 0.289 | 0.207 | 0.288 |
| | 96 | 192 | 0.239 | **0.295** | 0.237 | 0.324 | 0.248 | 0.319 | 0.377 | 0.411 | 0.283 | 0.342 | 0.307 | 0.369 | 0.269 | 0.324 | 0.291 | 0.351 |
| | 96 | 336 | **0.284** | **0.327** | 0.294 | 0.367 | 0.344 | 0.407 | 0.439 | 0.445 | 0.337 | 0.377 | 0.362 | 0.399 | 0.315 | 0.356 | 0.326 | 0.370 |
| | 96 | 720 | **0.346** | **0.370** | 0.388 | 0.421 | 0.465 | 0.494 | 0.629 | 0.569 | 0.416 | 0.426 | 0.459 | 0.454 | 0.385 | 0.391 | 0.409 | 0.420 |

Table 2: Comparing the VT (Vanilla Transformer), Informer, Autoformer and the FEDformer in our univariate and the multivariate settings. Values are bold if they are the best in the row and underlined if they are the best in the row for the model. The univariate setting outperforms its multivariate counterpart in the vast majority of all tests.

# 3 EXPERIMENTS

## 3.1 UNIVARIATE VS MULTIVARIATE

We test the four architectures in both the multivariate setting and our univariate setting. Both methods produce a forecast involving all the dimensions enabling direct comparisons to be made. The results can be found in Table 2. All models are trained to minimise the MSE and then evaluated on both MSE and MAE. We highlight the best overall loss in bold, and the best losses per model are underlined.

The VT, Informer and the Autoformer experience a significant improvement in forecasting accuracy when in our univariate setting. The results for the FEDformer are more mixed with neither mode having a clear advantage over the other. Overall the VT performs the best which is not unexpected due to it having the highest computational complexity of $O(l^2)$.

In Appendix C we test the VT in the traditional univariate setting, where a separate model is trained for each dimension, see Eq (2). We are only able to carry out this test for one architecture due to the high computation cost. The traffic dataset alone has 862 dimensions, see Table 1, requiring 862 models to be trained and tested, which took 3 weeks on a pair of Nvidia V100 GPUs.

The success of our univariate setting is surprising as it has less information to base the forecast on. The following experiments explore the reasons behind this finding.

## 3.2 INTER-DIMENSIONAL DEPENDENCIES

By visualising the impact each input dimension has on the dimensions within the forecast, we are able to identify whether the model has learnt inter-dimensional patterns. These will present themselves as an input dimension which affects multiple output dimensions.

For the VT, the Exchange dataset saw the largest improvement in the univariate setting and the Illness dataset was the only one that saw an improvement when in the multivariate setting. We visualise and compare the input output dimension interactions for the two datasets in Figure 1.

For Exchange, a strong diagonal pattern through the center is present. This indicates that each dimension is highly independent and the model has not learnt any inter-dimensional predictive patterns. In contrast, the visualisation for the Illness dataset shows multiple vertical lines. This shows that the model is drawing on multiple dimensions when predicting each output dimension; the forecast is based on inter-dimensional patterns. For example, dimension 3 is predictive of all dimensions, especially 0 and 1. Dimension 4, on the other hand, has nearly no predictive value.

The presence of predictive inter-dimensional patterns which the VT in the multivariate setting has learnt, explains why it outperform its univariate counterpart. However, in Appendix B we give the visualisations for all the datasets and it can be seen that Electricity and Traffic also have inter-dimensional patterns but the univariate setting still outperforms the other. These datasets have a much higher dimensional count, see Table 1, and in the next section we explore the impact of this.

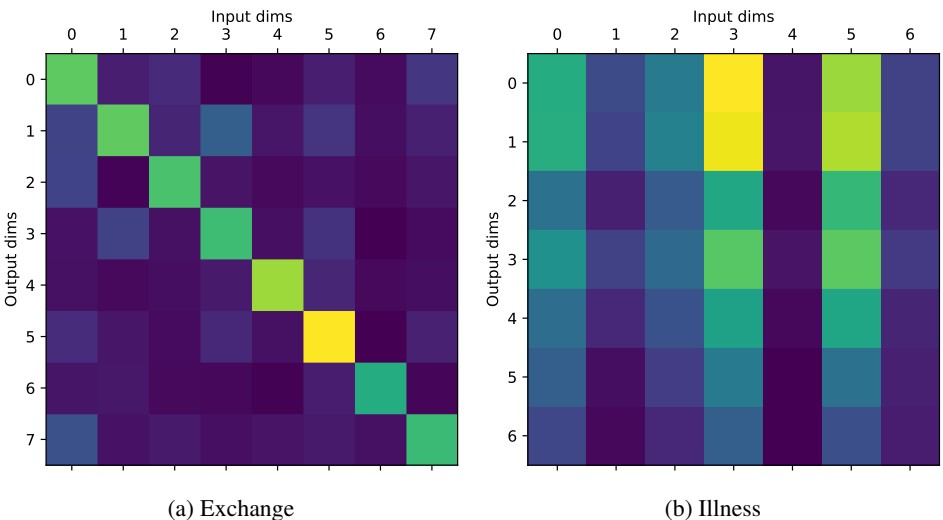

(a) Exchange                    (b) Illness

Figure 1: Here we visualise the importance of each input dimension, relative to each output dimension, given by a brighter colour. The dimensions with Exchange are showing to be highly independent, with no predictive inter-dimensional patterns being learnt. In contrast, the vertical columns of the Illness visualisation show that the model uses the information within specific dimensions to forecast multiple others. Visualisations for all the other datasets with the VT model can be found in Appendix B.

## 3.3 DIMENSION COUNT IMPACT

Our univariate setting outperforms the multivariate even when inter-dimensional patterns are present. By training the VT for a varying number of dimensions per dataset we are able to determine the precise impact of the dimension count, see Figure 2.

For 4 of the 6 datasets, there is a clear and strong positive correlation between the number of dimensions and the model's forecasting loss. Exchange has the strongest link. The model accuracy is significantly reduced as more dimensions are added.

A notable exception is the Illness dataset, which has a weak negative correlation meaning that the model benefits from access to the other dimensions and is likely learning inter-dimensional patterns.

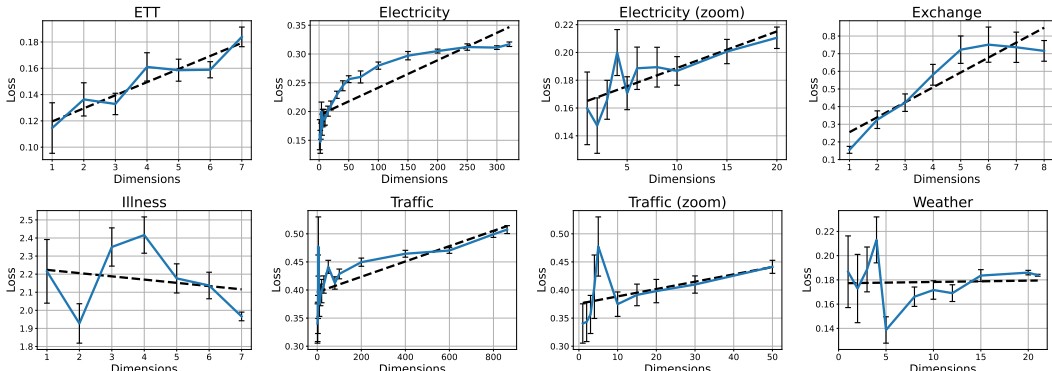

Figure 2: Here we explore how training VT models on datasets with a varying number of dimensions impacts forecasting accuracy. Random subsets of dimensions are selected from every dataset and a model is trained. This is repeated multiple times to calculate the standard error. The dashed line indicates the linear trend line, and for most datasets accuracy worsens as the model is trained on additional dimensions. The Illness dataset is the only one that sees an improvement when all dimension are available.

In Appendix A we give an alternative version of Figure 2 where each individual data point is given. We also explain why there is a generally higher variance for the first points of each plot.

In summary, Figure 2 shows that the VT model is impeded by high dimensional datasets and that this affect drowns out the benefits of inter-dimensional patterns for the Electricity and Traffic datasets.

## 3.4 DATASET SIZE REQUIREMENTS

Larger training datasets can often improve the performance of a model. We investigate whether either our univariate or the multivariate setting would be improved with additional data. We make the assumption that if the performance decreases with less data, then it will improve with more data. See Figure 3.

The data requirements for the multivariate setting is greater than that of the univariate setting. For Electricity, ETT, Traffic and Weather datasets, the univariate converges rapidly, whereas, the multivariate needs a larger amount of data, with Weather requiring the most (80% of the full training set). All these datasets appear to converge before a set size of 100% and so the performance of the multivariate setting is likely not limited by the amount of data.

The results for the Exchange and Illness datasets are more complex. For exchange, the univariate plot quickly improves and converges but the multivariate setting sees no improvement and is unstable suggesting that more data is required.

The Illness dataset gives the surprising pattern of an increasing loss with an increasing data size. This suggests that as more data is added to the training set, it becomes less representative of the test split, however, we are cautious to draw conclusions from this due to the exceptionally small size of the dataset. Illness has a times series length 70 times shorter than the longest, ETT, and Exchange is the second smallest dataset, see Table 1.

In summary, the multivariate setting requires larger datasets than the univariate, but the 4 largest datasets are large enough for this not to be a issue. The two smallest datasets, Exchange and Illness would appear to benefit from more information due to the unusual and unstable results.

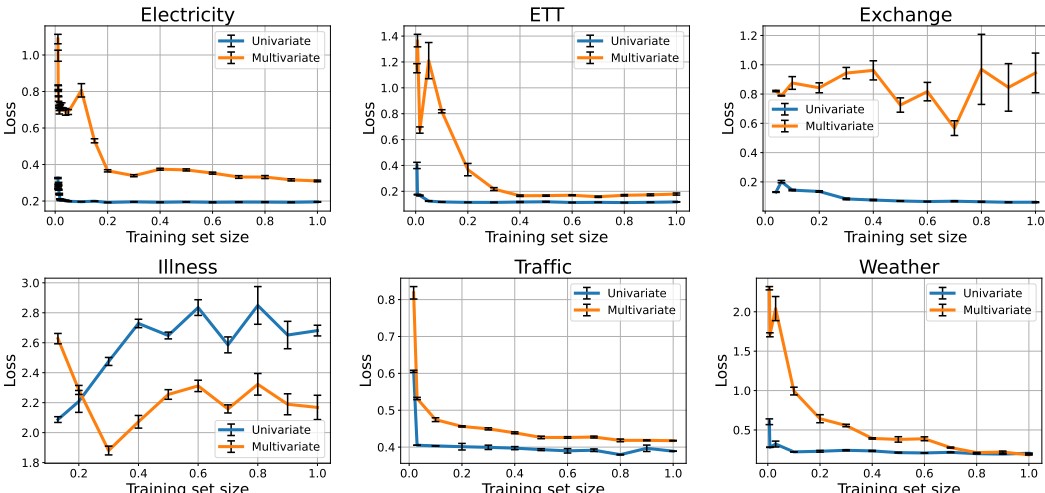

Figure 3: Evaluating model performance on reduced training set sizes. The loss is plotted against the training size, which is the fraction of the training data that is used. A values of 0.2 means that only the final 20% of the set is used. The error bars represent the standard error.

## 4 CONCLUSION

Our univariate setting outperforms its multivariate counterpart in the majority of test cases. This is due to the models being impeded by the additional dimensions in the multivariate setting, pointing to a flaw in the current Transformer based long-term TSF architectures. We expect that a form of variable selection is likely needed to avoid the noise from the dimensions crowding out useful inter-dimensional signals. Work from Lim et al. (2021) is relevant, however their mechanism selects relevant input features at each time step and does not address the issue of certain dimensions only being relevant to the forecast of other specific dimensions.

If a dataset contains inter-dimensional predictive patterns and has a low dimension count, the multivariate setting should perform best. In all other cases, we expect our univariate mode to achieve the best forecasting accuracy.

*Our code will be made publicly available on the completion of the review process.*

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

# APPENDIX

## A   DIMENSION COUNT IMPACT - SCATTER PLOTS

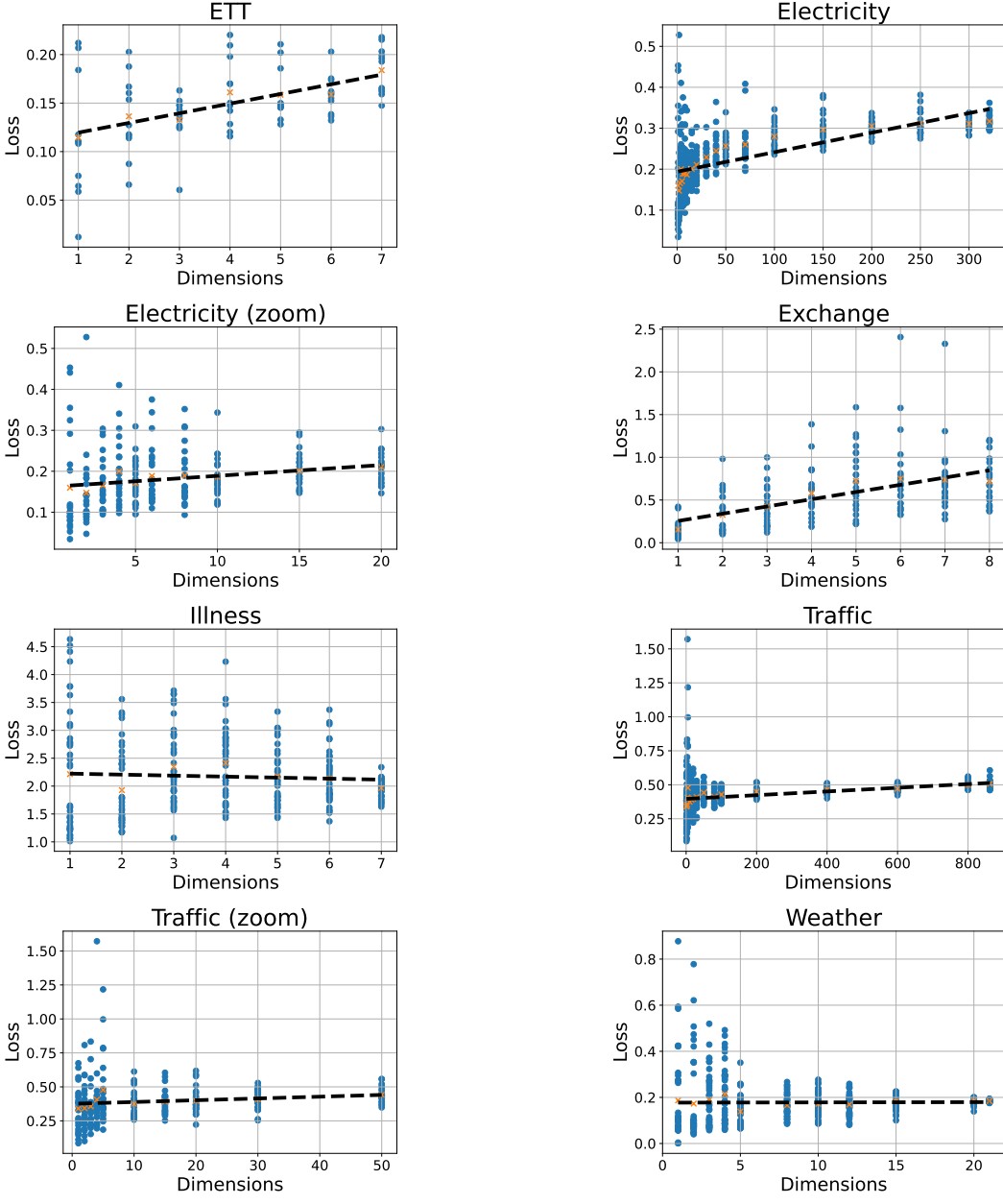

Figure 4: These plots are the same as in Figure 2, however, the individual data points are shown, instead of being averaged. There is a greater variance at the lower dimensions due to the high variation in how hard each dimension is to forecast. At higher dimension counts, this variance averages out.

# B UTILISATION OF MULTIVARIATE DATA

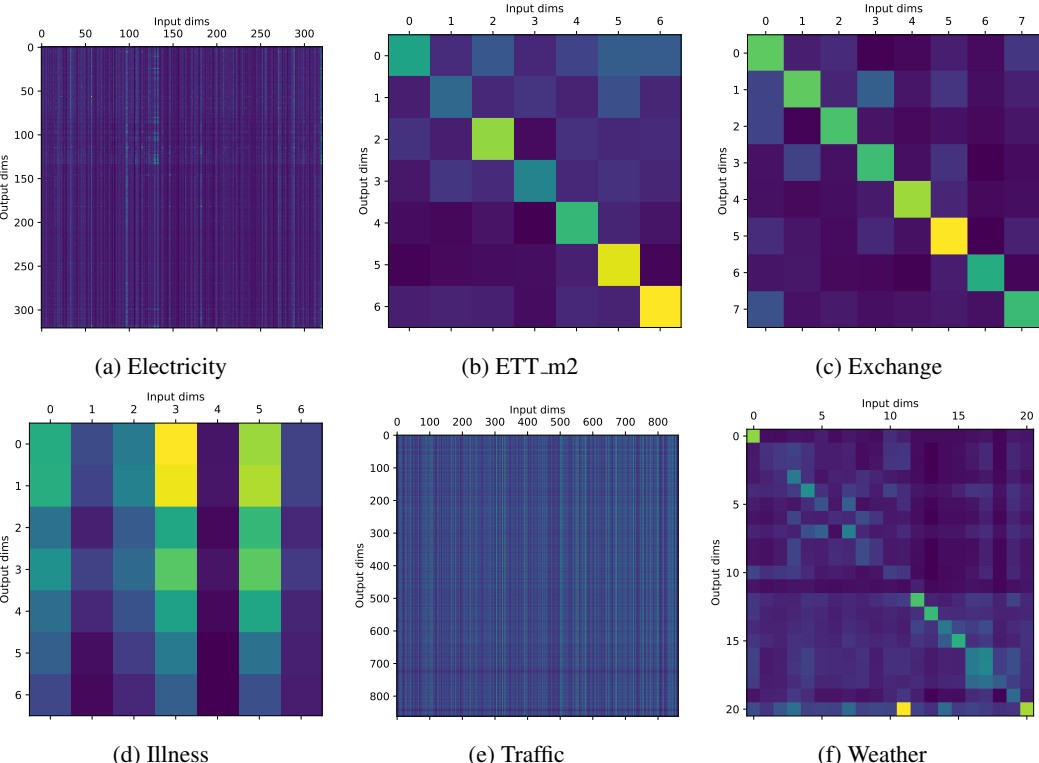

Figure 5: A visualisation of the importance of each input dimension in relation to each output dimension over all the datasets for the VT model. Brighter colours indicate a greater relevance.

# C  TRADITIONAL UNIVARIATE FORECASTING

| | History | Horizon | VT Trad Univariate MSE | MAE | VT Univariate MSE | MAE | VT Multivariate MSE | MAE | Informer Univariate MSE | MAE | Informer Multivariate MSE | MAE | Autoformer Univariate MSE | MAE | Autoformer Multivariate MSE | MAE | FEDformer Univariate MSE | MAE | FEDformer Multivariate MSE | MAE |
|---|---|---|---|---|---|---|---|---|---|---|---|---|---|---|---|---|---|---|---|---|
| Electricity | 96 | 96 | 0.175 | 0.271 | 0.194 | 0.280 | 0.311 | 0.410 | 0.154 | 0.248 | 0.282 | 0.377 | 0.196 | 0.288 | 0.202 | 0.316 | 0.213 | 0.300 | 0.196 | 0.312 |
| | 96 | 192 | 0.182 | 0.279 | 0.194 | 0.283 | 0.312 | 0.412 | 0.169 | 0.262 | 0.276 | 0.368 | 0.227 | 0.309 | 0.220 | 0.327 | 0.216 | 0.303 | 0.205 | 0.320 |
| | 96 | 336 | 0.199 | 0.296 | 0.207 | 0.297 | 0.316 | 0.414 | 0.196 | 0.293 | 0.285 | 0.376 | 0.237 | 0.321 | 0.224 | 0.334 | 0.224 | 0.317 | 0.216 | 0.331 |
| | 96 | 720 | 0.267 | 0.335 | 0.239 | 0.328 | 0.326 | 0.418 | 0.226 | 0.317 | 0.309 | 0.391 | 0.379 | 0.420 | 0.263 | 0.360 | 0.259 | 0.343 | 0.239 | 0.348 |
| ETT | 96 | 96 | 0.116 | 0.236 | 0.114 | 0.232 | 0.197 | 0.343 | 0.128 | 0.246 | 0.179 | 0.297 | 0.137 | 0.261 | 0.139 | 0.255 | 0.126 | 0.252 | 0.124 | 0.243 |
| | 96 | 192 | 0.154 | 0.267 | 0.143 | 0.264 | 0.225 | 0.340 | 0.158 | 0.280 | 0.221 | 0.336 | 0.169 | 0.292 | 0.192 | 0.297 | 0.159 | 0.282 | 0.150 | 0.267 |
| | 96 | 336 | 0.238 | 0.317 | 0.178 | 0.300 | 0.364 | 0.417 | 0.198 | 0.312 | 0.239 | 0.347 | 0.198 | 0.313 | 0.207 | 0.313 | 0.193 | 0.307 | 0.183 | 0.294 |
| | 96 | 720 | 0.398 | 0.393 | 0.230 | 0.346 | 0.881 | 0.575 | 0.265 | 0.369 | 0.438 | 0.480 | 0.261 | 0.360 | 0.264 | 0.359 | 0.252 | 0.352 | 0.244 | 0.339 |
| Exchange | 96 | 96 | 0.384 | 0.426 | 0.066 | 0.191 | 0.553 | 0.563 | 0.188 | 0.316 | 0.966 | 0.795 | 0.121 | 0.252 | 0.152 | 0.278 | 0.097 | 0.219 | 0.127 | 0.257 |
| | 96 | 192 | 0.700 | 0.612 | 0.119 | 0.259 | 1.320 | 0.912 | 0.320 | 0.433 | 1.228 | 0.878 | 0.244 | 0.360 | 0.402 | 0.459 | 0.200 | 0.324 | 0.274 | 0.379 |
| | 96 | 336 | 1.061 | 0.772 | 0.194 | 0.331 | 0.857 | 0.754 | 0.653 | 0.617 | 1.508 | 1.005 | 0.469 | 0.509 | 0.649 | 0.598 | 0.358 | 0.438 | 0.440 | 0.490 |
| | 96 | 720 | 1.766 | 1.005 | 0.741 | 0.661 | 1.833 | 1.106 | 1.236 | 0.872 | 2.468 | 1.309 | 2.123 | 1.158 | 1.408 | 0.885 | 0.973 | 0.755 | 1.232 | 0.846 |
| Illness | 36 | 24 | 2.144 | 1.028 | 2.630 | 1.121 | 2.060 | 1.010 | 4.394 | 1.357 | 5.088 | 1.490 | 3.084 | 1.141 | 3.508 | 1.256 | 2.553 | 0.988 | 2.723 | 1.129 |
| | 36 | 36 | 2.410 | 1.094 | 3.114 | 1.284 | 2.113 | 1.012 | 4.392 | 1.362 | 4.746 | 1.444 | 2.828 | 1.081 | 3.348 | 1.187 | 2.532 | 0.968 | 3.028 | 1.187 |
| | 36 | 48 | 2.513 | 1.098 | 3.094 | 1.283 | 2.598 | 1.118 | 4.131 | 1.319 | 4.642 | 1.440 | 2.907 | 1.096 | 3.176 | 1.160 | 2.889 | 1.078 | 2.873 | 1.088 |
| | 36 | 60 | 2.681 | 1.151 | 2.905 | 1.233 | 2.791 | 1.166 | 4.315 | 1.382 | 4.885 | 1.478 | 2.820 | 1.079 | 3.003 | 1.121 | 2.868 | 1.071 | 3.008 | 1.163 |
| Traffic | 96 | 96 | 0.369 | 0.321 | 0.391 | 0.339 | 0.428 | 0.382 | 0.544 | 0.257 | 0.673 | 0.369 | 0.547 | 0.318 | 0.639 | 0.395 | 0.540 | 0.377 | 0.586 | 0.367 |
| | 96 | 192 | 0.369 | 0.315 | 0.372 | 0.315 | 0.431 | 0.388 | 0.491 | 0.260 | 0.721 | 0.387 | 0.616 | 0.339 | 0.675 | 0.423 | 0.659 | 0.381 | 0.606 | 0.374 |
| | 96 | 336 | 0.378 | 0.319 | 0.379 | 0.317 | 0.454 | 0.399 | 0.483 | 0.269 | 0.716 | 0.391 | 0.590 | 0.335 | 0.634 | 0.391 | 0.659 | 0.377 | 0.624 | 0.387 |
| | 96 | 720 | 0.404 | 0.337 | 0.402 | 0.337 | 0.450 | 0.385 | 0.611 | 0.305 | 0.796 | 0.424 | 0.688 | 0.374 | 0.652 | 0.398 | 0.637 | 0.391 | 0.624 | 0.378 |
| Weather | 96 | 96 | 0.174 | 0.248 | 0.194 | 0.257 | 0.189 | 0.281 | 0.222 | 0.306 | 0.338 | 0.396 | 0.223 | 0.293 | 0.228 | 0.305 | 0.217 | 0.289 | 0.207 | 0.288 |
| | 96 | 192 | 0.212 | 0.283 | 0.239 | 0.295 | 0.237 | 0.324 | 0.248 | 0.319 | 0.377 | 0.411 | 0.283 | 0.342 | 0.307 | 0.369 | 0.269 | 0.324 | 0.291 | 0.351 |
| | 96 | 336 | 0.256 | 0.322 | 0.284 | 0.327 | 0.294 | 0.367 | 0.344 | 0.407 | 0.439 | 0.445 | 0.337 | 0.377 | 0.362 | 0.399 | 0.315 | 0.356 | 0.326 | 0.370 |
| | 96 | 720 | 0.339 | 0.381 | 0.346 | 0.370 | 0.388 | 0.421 | 0.465 | 0.494 | 0.629 | 0.569 | 0.416 | 0.426 | 0.459 | 0.454 | 0.385 | 0.391 | 0.409 | 0.420 |

Table 3: Here we include the traditional univariate setting, which involves training a separate model for each dimension, see Eq (2). We compare the setting to the multivariate and our univariate one. Likely due to the extra capacity of the $d$ models, the setting improves in the datasets that have a high number of dimensions. This method, however, is untenable due to the massive number of models that are required to be trained. For this reason we were only able to test with one architecture, VT. Values are bold if they are the best in the row and underlined if they are the best in the row for the model.

