# OpenReview forum: "Univariate vs Multivariate Time Series Forecasting with Transformers"
_ICLR.cc/2023/Conference — Submitted to ICLR 2023_

### Official Review · Reviewer_rqx4 · 2022-10-25

**Confidence:** 4
**Correctness:** 3
**Technical Novelty And Significance:** 3
**Empirical Novelty And Significance:** 3
**Recommendation:** 6

**Clarity, Quality, Novelty And Reproducibility:**

Overall the paper is written in good quality though several minor issues need to be addressed. The paper studied a novel finding and provided some interesting insights which help understand the model's behavior in forecasting multivariate time series.

**Strength And Weaknesses:**

Strengths
1. A novel problem is studied
2. The experiment design is comprehensive

Weaknesses:
1. Some model settings issues are not clearly explained in the paper. See questions below.

**Summary Of The Paper:**

This paper studied an interesting problem where the time series forecasting model may fail to benefit from the extra information in a multivariate setting.

**Summary Of The Review:**

This paper is a solid submission providing insights into pattern learning in multivariate time series forecasting. In particular, the authors design a series of experiments to explain the counterintuitive finding where the forecasting model performs better in a univariate setting. I have several questions listed below:

1. It's unclear why the authors don't use the traditional multivariate forecast setting with univariate models. Is it because of the computation cost?
2. How is one single univariate model trained on all the individual dimensions? Some details are needed here.
3. What are the loss functions used to train the model? A shape-aware loss function (e.g., DILATE) or point-wise distance-based loss functions (e.g., MSE, MAE, etc.)? The latter type of function may cause the model to fail to capture the shape of the time series, which I think could be another potential reason why the univariate setting sometimes outperforms the multivariate setting.

---

> ### Author Response · Authors · 2022-11-10
> **Thank you for your feedback**
>
> Thank you for your feedback. Here we address each of your questions:
>
> 1. We are unsure what you mean by _traditional multivariate forecasting setting with the univariate models_. If you are referring to the traditional univariate models then computational cost is indeed the restraint, which we mention in the caption of Table 3. A separate model is required for each input dimension and so 862 models would be needed for the Traffic dataset alone.
>
> 2. We have added to Section 2.3 paragraph 4 which we hope improves the clarity. The process is straightforward. During training, instead of sampling and forecasting from a single dimension, univariate sequences are taken from any dimension, at random, and a univariate forecast is produced. This is further illustrated with Equations (2) and (3). We are interested to know if you feel this is now clear to the reader.
>
> 3. All models are trained to minimise the MSE and then evaluated on both MSE and MAE. This is inline with the methodology of the Informer, Autoformer and FEDformer. We have now added these details to the paper, see Section 3.1, paragraph 1.
>
>
>     While using a shape aware loss function would improve the ”shape” of the forecast, the findings from the DILATE paper show that training with DILATE does not improve, and in some instances worsens, the MSE results. We choose MSE to train with as it is a common standard and used in the papers we compare to.

---

### Official Review · Reviewer_qsvN · 2022-10-25

**Confidence:** 2
**Correctness:** 3
**Technical Novelty And Significance:** 1
**Empirical Novelty And Significance:** 2
**Recommendation:** 5

**Clarity, Quality, Novelty And Reproducibility:**

Very clear, good reproducibility.

Weak novelty

**Strength And Weaknesses:**

**Strength**

Very simple and easy to understand approach... With nice performances

** Weaknesses**

The contribution is rather limited, the authors could have conducted more experiments

One of the remaining open question at the end of the reading is: are the different channels specific or generic in the considered multivariate dataset and, if they are specific, how does the univariate identify each of them ? [probably from the data history before the prediction].
The proposed interaction metric shows that the historical data are sufficient to predict the next step for 5 datasets out of 6... But we don't know if the historical data can be compensate by channel correlation.

legend (or caption details) is missing on Fig. 2

**Summary Of The Paper:**

The author propose to measure the quality of a simple univariate transformer on multi-variate problem without any specific agregative strategy. They show that this simple approach is empirically efficent on various tasks in different situations. The authors also try to measure the interaction between the channels during the forcasting process to understand the prediction. First, they simply quantify the impact of the number of channel by repeated random channel sampling experiments. Then they propose a gradient based method to quantify all single interactions.
Strong refererences are chosen to establish baseline performances on classical multivariate datasets and the comparison is interesting: multivariate models fail on data with low interactions between channels and succeed on Illness.
Then the authors propose to measure the data requirements of the different pipelines. Without surprise, the gap between simpler univariate approaches and more complex multivariate model increase when the dataset is reduced.

**Summary Of The Review:**

The message of the article is very clear and well illustrated but the contribution is rather limited.

It is a smart but limited series of experiments that may interest the ICLR community.

---

> ### Author Response · Authors · 2022-11-09
> **Clarifications**
>
> Thank you for your feedback, we have now amended the caption of Figure 2, listing the linear trend line.
>
> We are interested in what additional experiments you think would be useful. You mention _specific_ and _generic_ channels, but we are unsure what you mean by these terms. Also, could you elaborate on your point about how historical data could compensate the channel correlation, so we can follow up and conduct more experiments as you suggest.

---

> > ### Author Response · Authors · 2022-11-18
> > **Clarificiations**
> >
> > We regret that we have not been able to receive your clarification and suggestion on the additional experiments as there is now no time to address them.  By and large, we and other reviewers are confident that enough experiments have been conducted to thoroughly support our conclusion - current multivariate models handle multivariate data poorly.
> >
> > Our alternative method of producing multivariate forecasts with a univariate model achieves state of the art results. The three hypotheses that we explore provide valuable insights to the reader and will aid in the creation of future architectures.
> > 1. Larger datasets will not solve the issue raised in our paper.
> > 2. We show that many of the benchmark datasets do not have interdimensional predictive patterns. For such datasets, multivariate models should, at a minimum, match the accuracy of univariate models. Instead, we have shown that they underperform.
> > 3. As more dimensions are added to a dataset, the multivariate performance worsens in a clear and significant manner.
> >
> > We believe that our paper has opened a new avenue of research - effectively dealing with multivariate data.  This will require the development of new architectures, and a theoretical exploration into what we have empirically shown.

---

> > > ### Comment · Reviewer_qsvN · 2022-11-21
> > > **Clarifications**
> > >
> > > Currently, you apply a single predictor on all channel and measure the performance. Given the simplicity of the approach, we would like to see why it works so well. Inter dimensional dependancies are interesting but we wonder:
> > >
> > > - are the channels correlated?
> > > - are the channel correlated with a shift?
> > > - can we measure a similarity between the channel in the latent space?
> > > - does it exist a  group of similar channel and then some specific channel?
> > > - is the forcasting similar for the different channel?
> > >
> > > => If there are some positive answers, it means that the predictor should indeed be unique
> > > =>  If not, It means, we would have another series of question:
> > >
> > > - are the signals completely different?
> > > - do they depend on external context (that could be modeled)?
> > > - are the good performances due to the complexity of the estimator that adresses several different problems at the same time?
> > >
> > >
> > > As a summary: as I said before, I consider that this article is simple and consistent... But a little bit too simple: I think the authors should have provide more analysis to explain the perfromance.
> > >
> > > It is clear that it may interest the community but the contribution is rather limited.

---

> > > > ### Author Response · Authors · 2022-11-25
> > > > **Clarifications**
> > > >
> > > > We agree that it would be interesting to investigate the correlations between the channels. Particularly, if there are shifted correlations present, then it would be expected that the model would draw on these and improve the forecast.
> > > >
> > > > Instead of looking at statistical correlations in the raw data, we have looked into what correlations the model has learnt between the channels, shown in Figure 1. Taking this approach removes uncertainty. We can be sure when a model is benefitting from correlations/patterns that range across channels.
> > > >
> > > > From Figure 1 for the Illness dataset, we can see that in the multivariate setting, multiple channels are being used to produce forecasts. It therefore makes sense and is expected that the multivariate setting outperforms the univariate for this dataset.
> > > >
> > > > In datasets where the model does not learn correlations between channels, such as Exchange, the multivariate setting has no advantage over our univariate one. We would therefore expect the results between the two to be the same. The fact that they are not, and that our univariate method improves significantly, leads us to investigate the three hypothesis which we have mentioned previously.
> > > >
> > > > Measuring the similarity between the latent spaces would be interesting but we are unsure what useful information this would give us. We would expect that similar channels would have similar latent spaces.
> > > >
> > > > Using a single predictor makes for a fairer comparison since the multivariate setting also involves just a single model. We do test separate univariate models for each channel in table 3 which we refer to as the traditional univariate setting. Increasing the number of models in this way appears to improve results for datasets with many dimensions. This is probably due to the greater capacity for learning predictive patterns.
> > > >
> > > > We are confident in our conclusions that current SOTA multivariate architectures are poorly designed to handle multivariate datasets and we hope our paper encourages more work in the area.

---

### Official Review · Reviewer_tf8M · 2022-10-25

**Confidence:** 3
**Correctness:** 3
**Technical Novelty And Significance:** 2
**Empirical Novelty And Significance:** 2
**Recommendation:** 5

**Clarity, Quality, Novelty And Reproducibility:**

I found the paper is clearly written and easy to understand.


**Strength And Weaknesses:**

The proposed method is technically simple, but it seems to show good performance, compared to popular time-series forecasting models on the  benchmark datasets.


**Summary Of The Paper:**

The authors handles multivariate forecasting problem by utilizing a univariate forecasting model to predict individual dimensions of multivariate setting. They validate the proposed model with its baseline (Transformer-based model, Informer, Autoformer) on benchmark datasets and the experimental results show that the proposed model outperforms its baselines on some tasks.


**Summary Of The Review:**

The authors provides a good description about their motivation and proposed method, if technically correct, seems to address the targeted problem. Although the shown results are quite interesting (regression tasks) with a good motivation, it is a bit hard to think the proposed method is novel as the proposed method is just a simple application of transformer-based univariate forecasting model on multivariate setting.

---

> ### Author Response · Authors · 2022-11-09
> **Thank you for your feedback**
>
> Our method is indeed very simple, and we believe it holds a lot a value for the ICLR community. Our simple yet novel change to the model brings unexpected but significant performance improvements for 5 out of the 6 benchmark datasets. The experiments we have conducted help to explain why this occurs, and we hope that our work will benefit the creation of future architectures.
>
> Furthermore, we do believe our work is novel. To the best of our knowledge, no one else in the literature has produced multivariate forecasts with our method, which we have demonstrated outperforms the current state of the art. While our work focuses on an empirical study, we firmly believe that it may trigger further study on this new method from a theoretical perspective.

---

### Decision · Program_Chairs · 2023-01-20

**Decision:**

Reject

**Justification For Why Not Higher Score:**

The paper in its current form is below the acceptance threshold. Bring it above the threshold will require substantial revision after further research.


**Justification For Why Not Lower Score:**

N/A


**Metareview: Summary, Strengths And Weaknesses:**

This paper presents a simple method that allows univariate transformer-based time series forecasting models to be used for the multivariate setting. The efficacy of the proposed method is mainly demonstrated through experiments. The technical novelty of the proposed method is limited though. However, this might be acceptable if the paper could present a very in-depth analysis to justify why such a simple idea can work so effectively, making the paper scientifically more appealing to the ICLR community. This work may have potential for publication in the future if the paper could be improved along this direction.